# Self-Chained Image-Language Model for Video Localization and Question Answering

**Shoubin Yu    Jaemin Cho    Prateek Yadav    Mohit Bansal**
UNC Chapel Hill
{shoubin, jmincho, praty, mbansal}@cs.unc.edu

## Abstract

Recent studies have shown promising results on utilizing large pre-trained image-language models for video question answering. While these image-language models can efficiently bootstrap the representation learning of video-language models, they typically concatenate uniformly sampled video frames as visual inputs without explicit language-aware, temporal modeling. When only a portion of a video input is relevant to the language query, such uniform frame sampling can often lead to missing important visual cues. Although humans often find a video moment to focus on and rewind the moment to answer questions, training a query-aware video moment localizer often requires expensive annotations and high computational costs. To address this issue, we propose **Se**lf-Chained **Vi**deo **L**ocalization-**A**nswering (SeViLA), a novel framework that leverages a single image-language model (BLIP-2) to tackle both temporal keyframe localization and question answering on videos. SeViLA framework consists of two modules: Localizer and Answerer, where both are parameter-efficiently fine-tuned from BLIP-2. We propose two ways of chaining these modules for cascaded inference and self-refinement. First, in the forward chain, the Localizer finds multiple language-aware keyframes in a video, which the Answerer uses to predict the answer. Second, in the reverse chain, the Answerer generates keyframe pseudo-labels to refine the Localizer, alleviating the need for expensive video moment localization annotations. Our SeViLA framework outperforms several strong baselines/previous works on five challenging video question answering and event prediction benchmarks, and achieves the state-of-the-art in both fine-tuning (NExT-QA and STAR) and zero-shot (NExT-QA, STAR, How2QA, and VLEP) settings. We show a comprehensive analysis of our framework, including the impact of Localizer, comparisons of Localizer with other temporal localization models, pre-training/self-refinement of Localizer, and varying the number of keyframes.[1]

## 1  Introduction

The recent success of large-scale pre-trained language models [2, 7, 65] has led to a burst of multimodal vision-and-language models that can jointly understand visual (image/video) and language data [64, 6, 60]. However, due to higher computational and annotation costs, video-language models (video-LMs) are more challenging to scale in terms of model and data size than image-language models (image-LMs). Hence, recent studies have explored efficient training of video-LMs by leveraging pre-trained image-LMs [44, 14, 23, 85, 70, 31, 19, 82]. While such a warm-start strategy facilitates visual representation learning of video-LMs, they typically concatenate uniformly/randomly sampled video frames as visual inputs without explicit language-aware, temporal modeling. However, such a simple uniform/random sampling of frames can lead to losing important visual cues, resulting in the video-LMs focusing on frames that are unimportant/irrelevant to language queries [42].

---

[1]Our code and checkpoints are available at: https://github.com/Yui010206/SeViLA

37th Conference on Neural Information Processing Systems (NeurIPS 2023).

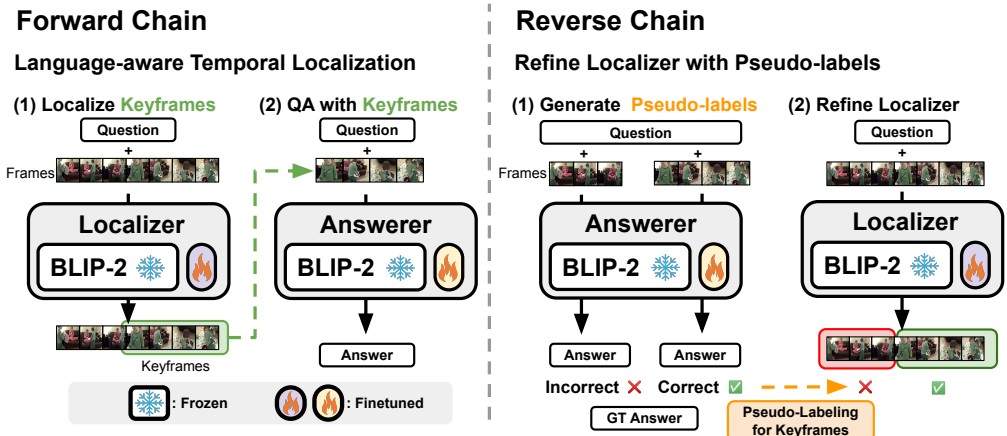

Figure 1: **Se**lf-**Ch**ained **Vi**deo **L**ocalization-**A**nswering (SeViLA) consists of a Localizer and an Answerer. **Left:** Forward chain for language-aware temporal keyframe localization and question answering. **Right:** Reverse chain for Localizer self-refinement with keyframe pseudo-labels.

To address this, we introduce **Se**lf-**Ch**ained **Vi**deo **L**ocalization-**A**nswering (SeViLA), a novel video-language framework where we adopt a single image-LM to handle both temporal localization and question answering on videos, while avoiding expensive language-aware, temporal grounding annotations (plus self-refinement [48] between the two modules). Our SeViLA framework obtains two modules, a **Localizer** and an **Answerer** through parameter-efficient fine-tuning [62] of BLIP-2 [35], a recent state-of-the-art image-LM. SeViLA tackles video-language tasks by chaining the output of Localizer to the input of Answerer (*forward chain*, Fig. 1 left), while the Answerer gives feedback to refine the Localizer (*backward chain*, Fig. 1 right). In the **forward chain**, Localizer leverages the original image-language understanding of the BLIP-2 backbone and chooses the important language-aware video keyframes via the localization prompt "Does the information within the frame provide the necessary details to accurately answer the given question?" for each video frame. Then Answerer takes the concatenation of selected keyframes as visual input to predict video-level answers. In the **backward chain**, we generate keyframe pseudo-labels [26] to refine the Localizer, where we label a video frame as a keyframe if Answerer can output the correct answer using that frame. This self-refinement improves the language-aware temporal localization accuracy and alleviates the need for expensive keyframe annotations.

We demonstrate the effectiveness of SeViLA framework on five challenging video question answering and event prediction benchmarks (NExT-QA, STAR, How2QA, TVQA, and VLEP) [75, 77, 36, 27, 28], where SeViLA outperforms several strong baselines/previous works, and achieves the state-of-the-art in both fine-tuning (NExT-QA and STAR) and zero-shot (NExT-QA, STAR, How2QA, and VLEP) settings. We also show that our Localizer can be used as a strong stand-alone moment retrieval model. We present a comprehensive analysis to elaborate the design choices of the proposed framework, including the impact of temporal localization, the impact of the self-refinement process (backward chain), and varying the number of keyframes. Our contributions are summarized as follows:

- A novel video-language framework SeViLA, where the Localizer and Answerer are initialized from a single image-language model to handle temporal localization and question answering on videos, respectively.
- A new self-refinement method for language-aware temporal keyframe localization, where the Answerer generates keyframe pseudo-labels to refine the Localizer, without expensive temporal grounding annotation.
- Strong empirical performance with state-of-the-art on multiple video-language benchmarks.
- Comprehensive analysis elaborating the design choices of the proposed framework.

## 2   Related Work

**Image-Language Pre-trained Models.** As the demand for cross-modal applications continues to grow, image-language pre-training studies have received tremendous attention and success. Image-

language pre-trained models [55, 12, 69, 5, 36, 64, 31, 34, 86] have advanced more rapidly than video-language pre-trained models [71, 90, 45, 17, 80, 67, 81, 83, 87, 37, 88], both in terms of model [61, 38, 35, 1, 70, 43] and pre-training data scale [61, 1, 89, 21] (more detailed model size and pre-training data scale comparisons are in Appendix). This can be attributed to the increased accessibility of image data and the comparatively simpler data structures, which makes scaling up image-language learning easier [58]. In our paper, SEVILA is built on the recent state-of-the-art image-LM BLIP-2 [35] and extends it to adopt video input for video-language tasks. We also compare our SEVILA framework with the current state-of-the-art video-LM, InternVideo [71], to demonstrate the superiority of a large image-LM that incorporates keyframe localization.

**Image-to-Video Transfer Learning.** The gap between image- and video-language models has inspired numerous useful methods focusing on image-to-video transfer learning, which leverage a limited number of video frames to enhance learning efficiency [80, 32, 23, 46, 15, 44, 14, 31, 4, 82, 68]. Luo et al. [44] adapt pre-trained CLIP [55] backbone to solve video clip retrieval. Yang et al. [85] extend frozen bidirectional language models [66] to incorporate multiple images and apply additional video-level pre-training to facilitate model adaptation [62]. Wang et al. [72] convert multiple images into hierarchical captions, arranged with a temporal order prompt to help language models comprehend video-level events. However, these works employ a uniform sampling strategy that is not language-aware. This can lead to the loss of key visual cues for temporal modeling and even burden the models with irrelevant information [31, 76]. In this paper, we propose a LOCALIZER to provide language-aware visual information to video-language tasks.

**Language-aware Keyframe Localization.** Many methods [42, 3, 18, 54, 24, 42, 41, 73, 9] have been proposed to address the challenge of language-aware keyframe localization. Buch et al. [3] optimized an end-to-end pipeline using answer labels to select a single keyframe for downstream tasks. Lu et al. [42] selects frames with separate image and language models, and answers questions by a QA model with multiple training objectives. Qian et al. [54] designed a video clip proposal model with predefined ranges, iteratively training it with a QA model. Kim et al. [24] utilized a semi-parametric retriever to obtain keyframes based on frame and language feature similarity. We adopt a large image-LM as our LOCALIZER and chain it with an ANSWERER. Our LOCALIZER can help to fine-tune ANSWERER in the forward chain and be refined with pseudo-labels in the reverse chain.

## 3 Method: SEVILA

In this section, we introduce the method details of our Self-Chained Video Localization-Answering (SEVILA) framework. First, we provide BLIP-2 preliminaries, which serve as the foundation for our framework. Then we elaborate our design of the BLIP-2 LOCALIZER and BLIP-2 ANSWERER for temporal localization and question answering on videos. Finally, we present the SEVILA framework's training and inference processes in the forward and reverse chain.

### 3.1 Preliminaries: BLIP-2

We adopt BLIP-2 [35] as the backbone of our SEVILA framework. BLIP-2 is a recent state-of-the-art pre-trained image-language model (image-LM) comprising of: (1) a frozen image encoder [11, 16]; (2) a frozen large language model (LLM) [7, 91]; and (3) a Q-former, which is a trainable transformer [66] module that bridges the image encoder and LLM, similar to acting as an adapter [62, 20]. It takes as input visual features $h$ from the image encoder and learnable query embeddings $q$, and outputs fixed-length visual features $v$. The BLIP-2 Q-Former undergoes a two-stage pre-training. First, it connects to the image encoder to perform image-to-text pre-training. This stage enables the Q-Former to extract the most informative visual information for the text and remove any irrelevant details in $v$. Subsequently, the Q-Former is connected to the LLM to leverage its generative language capabilities. This is achieved using a fully-connected layer to project query embeddings into the LLM's dimension with image-to-text pre-training. As a result, these query features serve as soft visual prompts [22] for the LLM. With the two-stage pre-trained Q-former and LLM, BLIP-2 shows advanced performance on various image-language tasks. In our SEVILA framework, we adopt BLIP-2 as the basic building block for both video temporal localization and question answering modules. We retain the visual encoder and the LLM from BLIP-2 by keeping them frozen during training. In this case, only these two Q-formers are updated during the LOCALIZER and ANSWERER training (see Sec. 3.3).

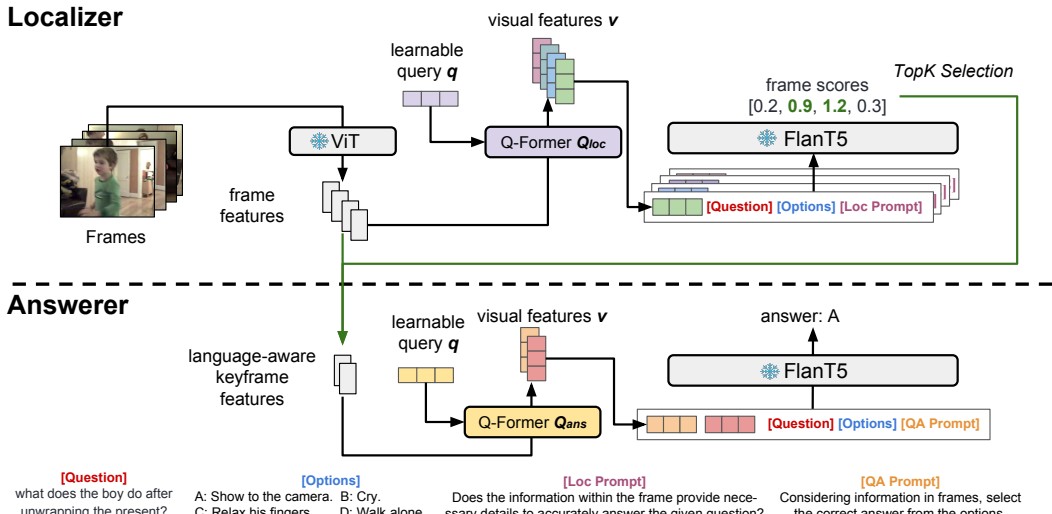

Figure 2: In SEVILA framework, LOCALIZER (top) selects top-K video frames, which guides ANSWERER (bottom) to focus on important language-aware video moments and predict answers. Both LOCALIZER and ANSWERER are initialized from a single pre-trained BLIP-2 model, where only Q-formers and a linear layer (2.5% of total parameters) are tuned for each module. We omit the linear layer after the Q-former for simplicity.

## 3.2 Self-Chained Video Localization-Answering

**Adapting BLIP-2 to Temporal Localization and Question Answering on Videos.** As illustrated in Fig. 2, our SEVILA framework adopts BLIP-2 to tackle both video temporal localization and question-answering. We assign BLIP-2 two roles of being a LOCALIZER and an ANSWERER by using different Q-formers. We first elaborate our LOCALIZER and ANSWERER in detail as follows:

**LOCALIZER.** We first extract frame features via the frozen image-encoder ViT [16], referred to $E_v$. Given the video, we uniformly sample $n$ frames $\{f_1, ..., f_n\}$. We then obtain $i_{th}$ frame feature $h_i$ as $h_i = E_v(f_i)$. Finally, we represent the video as a set of frame features $V = \{h_1, ..., h_n\}$. These features are extracted once and then saved to be subsequently reused by the LOCALIZER and the ANSWERER. The primary objective of the LOCALIZER is to select $k$ language-aware keyframe features from $V$, where $k$ is typically much smaller than $n$. As illustrated in Fig. 2 (top), we then independently extract visual query features $v_i$ from original frame features in $V$ via a Q-Former $Q_{loc}$. Next, visual query features $v_i$ and language contexts $L$ are concatenated and fed into the LLM (Flan-T5 [7]), where we create $L$ by combining question, options, and localization prompt "Does the information within the frame provide the necessary details to accurately answer the given question?". The LOCALIZER outputs the score for each frame $s_i$, which is the probability of generating a word 'yes', given the visual features $v_i$ and language context $L$: $s_i = LLM(concat(v_i, L))$. We can localize language-aware keyframes $K = \{v_1^k, ..., v_K^k\}$, based on the top-k frame scores. Our LOCALIZER can be formulated as:

$$K = \text{LOCALIZER}(V, L), \quad |K| = k \ll n \tag{1}$$

**ANSWERER.** With the keyframe set $K$ obtained from the LOCALIZER, as illustrated in Fig. 2 (bottom), we can proceed to generate answers using the ANSWERER. We first obtain keyframe query features $v_i$ by processing them through $Q_{ans}$, following the same procedure used in the LOCALIZER. Next, we feed the LLM with all query features and language contexts by concatenating them and obtain the video-level answer $a = LLM(concat(v_1^k, ..., v_K^k, L))$[2]. Then, the frozen LLM can conduct temporal modeling with multiple frame inputs. Our ANSWERER can be formulated as:

$$a = \text{ANSWERER}(K, L) \tag{2}$$

---

[2]We also insert a frame ID token before each frame, where we omit the notations from the equation for simplicity.

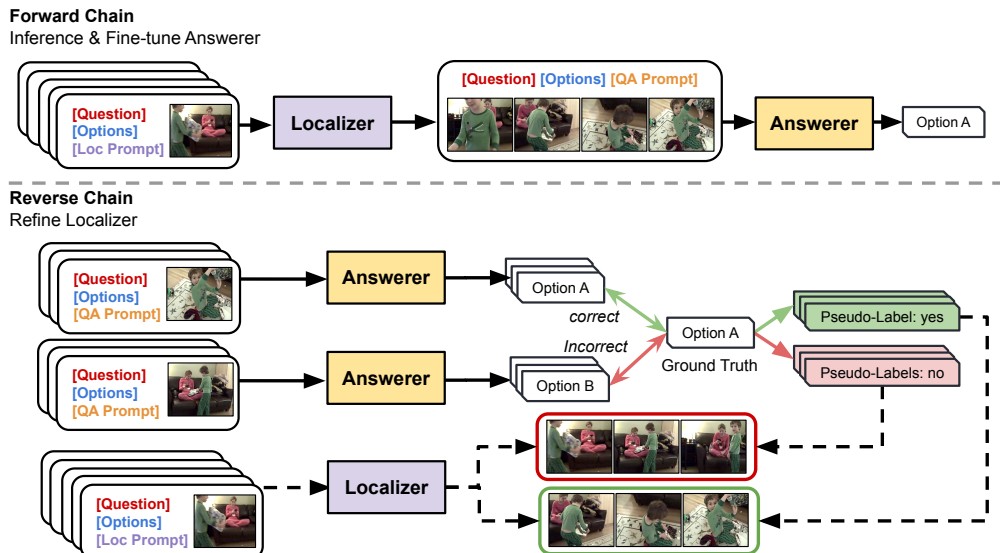

Figure 3: **Top:** In the forward chain, the LOCALIZER finds multiple language-aware keyframes, then the ANSWERER utilizes these keyframes to predict answers. We use the forward chain for both inference and ANSWERER fine-tuning. **Bottom:** In the reverse chain, we generate keyframe pseudo-labels by using the ANSWERER to refine the LOCALIZER.

## 3.3 Training ANSWERER and LOCALIZER via Self-Chaining

**Fine-tuning ANSWERER in forward chain.** As illustrated in Fig. 3 (top), we fine-tune the ANSWERER on downstream tasks using keyframes from LOCALIZER via the forward chain. ANSWERER takes the keyframes generated by LOCALIZER. We compare the default setting to other settings (*e.g.*, input frames are uniformly selected) in Appendix.

**Refining LOCALIZER in reverse chain.** We adopt pseudo-labeling [26] in our reverse chain to address the costly frame-level localization annotations. We use binary pseudo-labels, where we label a video frame as a keyframe if ANSWERER can output the correct answer using that frame. As shown in Fig. 3 (bottom), The frozen ANSWERER is first prompted by a QA task prompt and generates a frame-level answer, then we obtain pseudo labels by comparing this prediction with the ground-truth answer. The LOCALIZER is trained to locate language-aware pseudo-label keyframes.

**Pre-training LOCALIZER with moment retrieval label.** To enhance our LOCALIZER, we conduct transfer learning from a video moment retrieval/grounding task [56, 30] via pre-training. We use videos, queries, and video-level temporal span labels from QVHighlights [30], and assign a binary localization label to each frame by comparing its timestamp to the span annotations. We provide more details of pre-training in Appendix.

## 4 Experiments

In this section, we first outline our experimental setup (Sec. 4.1). Then, we demonstrate the superiority of SEVILA framework on 5 challenging long-form video question answering and event prediction benchmarks in both fine-tuning (Sec. 4.2) and zero-shot (Sec. 4.3) settings. we also conduct ablation studies on SEVILA framework to show the effectiveness of each of its components on downstream tasks (Sec. 4.4). Next, We report the performance of our LOCALIZER on video moment retrieval (Sec. 4.5). Lastly, we perform in-depth quantitative, and qualitative analyses on our LOCALIZER to show the effect of our design for temporal keyframe localization (Sec. 4.6 and Appendix). More results on single v.s. multi-frame LOCALIZER, pre-training strategies, iterative self-refinement, computational cost, and extension to another Image-LM model are in Appendix.

## 4.1 Experimental setup

**Benchmarks.** We evaluate our SEVILA framework on 3 video-language tasks, including multi-choice Video Question Answering (**NExT-QA** [77], **STAR** [75], **How2QA** [36], **TVQA** [27]), Video Event Prediction (**VLEP** [28]), and Moment Retrieval (**QVHighlights** [30]). See Appendix for details.

Table 1: Fine-tuning results on video question answering (NExT-QA, STAR, How2QA, TVQA) and video event prediction (VLEP). We gray out the methods take extra speech input or use dense frames. We **bold** the best numbers, and underlined the second-best numbers. dense/1fps: the model takes dense (1fps) video frames instead of a fixed number of frames. $32 \rightarrow 4$: our Localizer selects 4 keyframes from 32 frames. * represents the results tested by ourselves. SeViLA[†] uses the zero-shot Localizer without refining on pseudo-labels via the reverse chain.

| Model (# Frames) | NExT-QA | | | | STAR | | | | | How2QA | TVQA | VLEP |
|---|---|---|---|---|---|---|---|---|---|---|---|---|
| | Tem. | Cau. | Des. | Avg. | Int. | Seq. | Pre. | Fea. | Avg. | | | |
| *(w/ speech input or use dense frames)* | | | | | | | | | | | | |
| HERO (dense/1fps) [36] | - | - | - | - | - | - | - | - | - | 73.8 | 73.6 | - |
| JustAsk (20) [84] | 51.4 | 49.6 | 63.1 | 52.3 | - | - | - | - | - | 84.4 | - | - |
| FrozenBiLM (10) [85] | - | - | - | - | - | - | - | - | - | 86.7 | 82.0 | - |
| VidIL 4-shot (12) [72] | - | - | - | - | - | - | - | - | - | - | - | 72.0 |
| T+T (dense/1fps) [40] | - | - | - | - | - | - | - | - | - | 92.4 | - | - |
| T+T (+ASR, dense/1fps) [40] | - | - | - | - | - | - | - | - | - | 93.2 | - | - |
| Flamingo-80B 32-shot (30) [1] | - | - | - | - | - | - | - | - | 42.2 | - | - | - |
| FrozenBiLM (10) [85] | - | - | - | - | - | - | - | - | - | 81.5 | 57.5 | - |
| All-in-One (32) [67] | 48.6 | 48.0 | 63.2 | 50.6 | 47.5 | 50.8 | 47.7 | 44.0 | 47.5 | - | - | - |
| Temp[ATP] (32) [3] | 49.3 | 48.6 | 65.0 | 51.5 | 50.6 | 52.8 | 49.3 | 40.6 | 48.3 | - | - | - |
| VGT (32) [78] | 55.0 | 52.2 | 64.0 | 55.0 | - | - | - | - | 44.2 | - | - | - |
| MIST (32) [18] | 56.6 | 54.6 | 66.9 | 57.1 | 55.5 | 54.2 | 54.2 | 44.4 | 51.1 | - | - | - |
| VFC (32) [50] | 53.3 | 57.6 | 72.8 | 58.6 | - | - | - | - | - | - | - | - |
| CoVGT (32) [79] | 57.4 | 58.8 | 69.3 | 60.0 | - | - | - | - | 45.9 | - | - | - |
| SeViT_FiD (10) [24] | - | - | - | 60.6 | - | - | - | - | - | - | - | - |
| HiTeA (16) [87] | 58.3 | 62.4 | 75.6 | 63.1 | - | - | - | - | - | - | - | - |
| InternVideo* (8) [71] | 58.5 | 62.5 | 75.8 | 63.2 | 62.7 | 65.6 | 54.9 | 51.9 | 58.7 | 79.0 | 57.2 | 63.9 |
| BLIP-2[voting] (4) | 65.2 | 70.1 | 80.1 | 70.1 | 52.3 | 54.8 | 49.0 | 51.2 | 51.8 | 79.6 | 54.5 | 67.0 |
| BLIP-2[concat] (Answerer) (4) | 68.1 | 72.9 | 81.2 | 72.6 | **65.4** | 69.0 | 59.7 | 54.2 | 62.0 | 82.2 | 59.8 | 68.6 |
| SeViLA[†] ($32 \rightarrow 4$) | 68.8 | 73.4 | **83.5** | 73.4 | 63.2 | 66.6 | 61.3 | 60.0 | 62.7 | **83.7** | 59.7 | **69.0** |
| SeViLA ($32 \rightarrow 4$) | **69.4** | **74.2** | 81.3 | **73.8** | 63.7 | **70.4** | **63.1** | **62.4** | **64.9** | 83.6 | **61.6** | 68.9 |

**Baselines.** We compare our SeViLA framework with the state-of-the-art video-language pre-trained model, InternVideo [71] as well as our backbone BLIP-2 [35]. We extend BLIP-2 to adapt videos in two settings: (1) BLIP-2[voting], which processes each uniformly sampled frame independently and obtains the final answer by majority voting on all frame-level answers, and (2) BLIP-2[concat], where Q-former processes each frame and Flan-T5 takes the concatenation of visual features as a prefix. BLIP-2[concat] is equivalent to our Answerer with uniformly sampled frames. See Appendix for details.

**Implementation Details.** SeViLA framework adopts BLIP-2 [35], an image-language model with 4.1B parameters and pre-trained on 129M images in total, including COCO [39], Visual Genome [25], CC12M [59], SBU [52], and 115M images from LAION400M [57]. See Appendix for details.

## 4.2 Fine-tuning Comparison to SOTA on Video QA and Event Prediction

We compare our SeViLA framework to recent state-of-the-art models on 4 video QA benchmarks and 1 video event prediction dataset. We show results in Table 1, and summarize our findings as follows.

**(a) Temporal modeling matters.** BLIP-2[voting] underperforms our BLIP-2[concat] (Answerer) and other video-LM models on STAR, How2QA, TVQA, and VLEP. Especially on STAR-Sequence, a task demanding heavy temporal understanding, our BLIP-2[concat] (Answerer) outperforms BLIP-2[voting] significantly by 13.1% (69.0% vs. 54.8%). As BLIP-2[voting] processes frames independently and lacks temporal modeling among frames, the result indicates that temporal modeling is essential to tackle video-language tasks and the effectiveness of our temporal modeling design.

**(b) Keyframe selection helps.** Our SeViLA[†] framework, featuring a zero-shot Localizer, leads on all tasks with an average advantage of 5.3% over the top video-LM (InternVideo). It also surpasses BLIP-2[concat] (Answerer) that uses uniform frame sampling on NeXT-QA (+1.2%), STAR (+0.7%), How2QA (+1.5%), and VLEP (+0.4%). This highlights the importance of keyframe selection in video-language tasks, even when using a zero-shot Localizer.

Table 2: Zero-shot results on video question answering and video event prediction.

| Model (# Frames) | NExT-QA | | | | STAR | | | | | How2QA | TVQA | VLEP |
|---|---|---|---|---|---|---|---|---|---|---|---|---|
| | Tem. | Cau. | Des. | Avg. | Int. | Seq. | Pre. | Fea. | Avg. | | | |
| *(w/ speech input or use dense frames)* | | | | | | | | | | | | |
| JustAsk (20) [84] | - | - | - | - | - | - | - | - | - | 51.1 | - | - |
| FrozenBiLM (10) [85] | - | - | - | - | - | - | - | - | - | 58.4 | 59.2 | - |
| ViperGPT (dense/1fps) [63] | - | - | - | 60.0 | - | - | - | - | - | - | - | - |
| Flamingo-80B (30) [1] | - | - | - | - | - | - | - | - | 39.7 | - | - | - |
| FrozenBiLM (10) [85] | - | - | - | - | - | - | - | - | - | 41.9 | 29.7 | - |
| VFC (32) [50] | 45.4 | 51.6 | 64.1 | 51.5 | - | - | - | - | - | - | - | - |
| InternVideo* (8) [71] | 43.4 | 48.0 | 65.1 | 49.1 | 43.8 | 43.2 | 42.3 | 37.4 | 41.6 | 62.2 | 35.9 | 58.7 |
| BLIP-2$^{\text{voting}}$ (4) | 59.1 | 61.3 | 74.9 | 62.7 | 41.8 | 39.7 | 40.2 | 39.5 | 40.3 | 69.8 | 35.7 | 63.8 |
| BLIP-2$^{\text{concat}}$ (Answerer) (4) | 59.7 | 60.8 | 73.8 | 62.4 | 45.5 | 41.8 | 41.8 | 40.0 | 42.2 | 70.8 | 36.6 | 64.0 |
| SeViLA$^{\dagger}$ (32→4) | **61.3** | **61.5** | **75.6** | **63.6** | **48.3** | **45.0** | **44.4** | **40.8** | **44.6** | **72.3** | **38.2** | **64.4** |

**(c) Self-refinement improves temporal localization.** For SeViLA, we refine the Localizer with pseudo-labels (see Sec. 3.3). Compared with SeViLA$^{\dagger}$, SeViLA further increases performance on NExT-QA (0.4%), STAR (+2.2%), and TVQA (+1.9%). SeViLA framework achieves new **state-of-the-art** fine-tuning performance on NExT-QA, STAR, TVQA, and VLEP, using only visual and language modalities. This illustrates the significance of temporal localization and the efficacy of our self-refinement method for keyframe localization.

### 4.3  Zero-shot Comparison to SOTA on Video QA and Event Prediction

We further compare our SeViLA framework to recent state-of-the-art models in the zero-shot setting. We show the zero-shot results in Table 2, then discuss the findings in the following.

**(a) Image-LM outperforms Video-LM, without video pretraining.** Surprisingly, BLIP-2$^{\text{voting}}$, without inter-frame temporal modeling, outperforms the previous state-of-the-art video-LM, InternVideo on several datasets that require temporal reasoning. BLIP-2$^{\text{voting}}$ outperforms InternVideo on NExT-QA (+13.6%), How2QA (+7.6%), and VLEP (+5.1%). On How2QA, BLIP-2$^{\text{voting}}$ even surpasses FrozenBiLM which performs extra speech and video pre-training by 11.4%. It highlights the potential of image-LM for video-language tasks due to its model size and sufficient pre-training.

**(b) Keyframe selection is more effective than uniform sampling.** Our SeViLA† framework, combining the zero-shot Localizer and the zero-shot Answerer, outperforms the BLIP-2$^{\text{concat}}$ (Answerer) with uniformly sampled frames on NExT-QA (+1.2%), STAR (+2.4%), How2QA (+1.5%), TVQA (+1.6%), and VLEP (+0.4%), achieving new **state-of-the-art** zero-shot performance on NExT-QA, STAR, How2QA, and VLEP, and new state-of-the-art on TVQA with only visual and language modalities. On STAR, our SeViLA$^{\dagger}$ framework even outperforms zero-shot Flamingo [1] with 80B parameters by 4.9%. The result demonstrates the effectiveness of our SeViLA framework to adapt video-language tasks and the importance of language-aware keyframe selection.

### 4.4  Ablation Studies on SeViLA Framework

We conduct ablation studies on our SeViLA framework about the effectiveness of Localizer and Answerer. We show the results in Table 3. We summarize our findings as follows:

**(a) Sparse frames outperform dense frames:** We observe performance declines when the Answerer adopts more frames (A *v.s.* B), confirming that sparse frames work better due to the original Image-LM model's limited temporal modeling ability, while too dense frames may distract the model.

**(b) Keyframes outperform uniformly sampled frames:** We compare Answerer with Localizer (SeViLA framework) and Answerer that takes uniformly sampled frames. We observed significant performance gains in the zero-shot Answerer setting when utilizing the zero-shot Localizer (B *v.s.* C), with improvements on NExT-QA (+1.0%), STAR (+2.4%), How2QA (+1.5%), TVQA (+1.6%), and VLEP (+0.4%) . And pseudo-label refinement for Localizer further increased performance by an average of 2.1% across all tasks (B *v.s.* D). In the fine-tuned Answerer setting, the benefits of the Localizer remained evident. Our SeViLA framework, which utilized keyframes from the

Table 3: Ablation studies on SEVILA framework. 'uniform' refers to the uniform sampling of video frames. LOCALIZER† refers to the zero-shot LOCALIZER without refining on pseudo-labels.

| | ANSWERER | | Keyframe | NExT-QA | | | | STAR | | | | | How2QA | TVQA | VLEP |
|---|---|---|---|---|---|---|---|---|---|---|---|---|---|---|---|
| | # frame | finetuned? | | Tem. | Cau. | Des. | Avg. | Int. | Seq. | Pre. | Fea. | Avg. | | | |
| A. | 32 | ✗ | uniform | 54.7 | 56.7 | 67.8 | 57.7 | 46.2 | 43.6 | 40.7 | 41.0 | 42.8 | 67.0 | 33.2 | 54.0 |
| B. | 4 | ✗ | uniform | 59.7 | 60.8 | 73.8 | 62.4 | 45.5 | 41.8 | 41.8 | 40.0 | 42.2 | 70.8 | 36.6 | 64.0 |
| C. | 4 | ✗ | LOCALIZER† | 61.3 | 61.5 | 75.6 | 63.6 | 48.3 | 45.0 | 44.4 | 40.8 | 44.6 | 72.3 | 38.2 | 64.4 |
| D. | 4 | ✗ | LOCALIZER | 62.3 | 63.1 | 74.9 | 64.6 | 49.0 | 46.4 | 45.2 | 41.6 | 45.5 | 72.9 | 39.1 | 64.6 |
| E. | 4 | ✓ | uniform | 68.1 | 72.9 | 81.2 | 72.6 | 65.4 | 69.0 | 59.7 | 54.2 | 62.0 | 82.2 | 59.8 | 68.6 |
| F. | 4 | ✓ | LOCALIZER† | 68.8 | 73.4 | 83.5 | 73.4 | 63.2 | 66.6 | 61.3 | 60.0 | 62.7 | 83.7 | 59.7 | 69.0 |
| G. | 4 | ✓ | LOCALIZER | 69.4 | 74.2 | 81.3 | 73.8 | 63.7 | 70.4 | 63.1 | 62.4 | 64.9 | 83.6 | 61.6 | 68.9 |

Table 4: Comparison on QVHighlights test split. We aggregate frame-level results of our LOCALIZER for video-level evaluation (see Appendix).

| Model | R1@0.5 | R1@0.7 | mAP |
|---|---|---|---|
| CAL [13] | 25.4 | 11.5 | 9.8 |
| XML [29] | 41.8 | 30.3 | 32.1 |
| Moment-DETR [30] | 52.8 | 33.0 | 30.7 |
| QD-DETR [51] | 62.4 | 44.9 | 39.8 |
| LOCALIZER (Ours) | 54.5 | 36.5 | 32.3 |

Table 5: The impact of QVHighlights Pre-Training (PT) and Self-Refinement (SR) for our LOCALIZER in Sec. 3.3.

| PT | SR | NExT-QA | | | | How2QA |
|---|---|---|---|---|---|---|
| | | Tem. | Cau. | Des. | Avg. | |
| - | - | 60.4 | 61.0 | 74.6 | 62.9 | 70.7 |
| ✓ | - | 61.3 | 61.5 | 75.6 | 63.6 | 72.3 |
| - | ✓ | 62.1 | 62.6 | 75.1 | 64.3 | 72.8 |
| ✓ | ✓ | 62.3 | 63.1 | 74.9 | 64.6 | 72.9 |

LOCALIZER, outperformed the 4-frame ANSWERER by an average of 0.7% across tasks (E *v.s.* F). Pseudo-label refinement continues to be effective in this setting, providing an average boost of 1.5% across all tasks (E *v.s.* G). These results demonstrate that keyframe selection contributes to non-trivial improvements in video-language tasks for zero-shot and fine-tuning settings.

## 4.5 Comparison to State-of-the-Art on Video Moment Retrieval

In this section, we evaluate our LOCALIZER on the video moment retrieval task. We pre-train the LOCALIZER on the QVHighlights [30] dataset, as discussed in Sec. 3.3, and then assess its performance on the same dataset. To test on QVHighlights, we first extract video frames at 0.5 fps, following Moment-DETR [30] and pass them through our LOCALIZER to obtain binary frame-level predictions that indicate whether a frame matches the query sentence. Next, we combine these predictions into video-level temporal span predictions. We aggregate frame-level predictions into video-level spans by merging adjacent positive predictions with intervals not exceeding a threshold. These merged results are then consolidated into a single video-level span. More information on the aggregation process can be found in Appendix. Interestingly, as shown in Table 4, our LOCALIZER, which has no temporal modeling/training and operates on frame-level, outperforms many previous methods [13, 30, 29] with complex temporal modeling and video data training. It demonstrates our LOCALIZER can further work as a standalone model for certain tasks. It also suggests large image-LM with temporal designs may be a promising study for video moment retrieval. This is evidenced by our LOCALIZER's superior performance compared to Moment-DETR, despite the absence of temporal modeling.

## 4.6 Detailed Analysis on the Localizer

In this section, we first analyze the impact of pre-training and self-refinement on our LOCALIZER. Then, we compare our LOCALIZER with other keyframe selection methods in both zero-shot and fine-tuning settings. Next, we experiment with different keyframe selection ranges and quantities in our LOCALIZER to assess the impact of temporal localization on the overall performance. We further show the impact of LOCALIZER in ANSWERER fine-tuning. We also present upper-bound analysis based on BLIP-2 and oracle keyframe localization. Lastly, we provide visualization results of our LOCALIZER. Additional experiments are in Appendix.

**Ablation on LOCALIZER pre-training and self-refinement.** We performed these ablation studies with the zero-shot 4-frame ANSWERER. As shown in Table 5, the untrained BLIP-2 LOCALIZER

results in only a minor improvement to the ANSWERER. Moreover, both QVHighlights pre-training and self-refinement via the reverse chain independently provide significant performance boosts. The optimal results are achieved when both pre-training and self-refinement are applied. It further demonstrates our method is label-efficient for keyframe temporal localization.

**Comparison with other keyframe selection methods.** In Table 6, we compare our LOCALIZER with different keyframe localization methods, including CLIP [55], Moment-DETR [30] which are zero-shot, and ATP [3], Differentiable Top-K [8] which are fine-tuned with answer-label. We combine those keyframe localization methods with our zero-shot ANSWERER. Those methods select 4 keyframes from 32 uniformly sampled frames. We find that keyframes from neither CLIP nor Moment-DETR can not help ANSWERER. It may be due to their CLIP pre-training on images and short declarative sentences, which fail to produce question-aware visual features, and distract ANSWERER with irrelevant features. In contrast, our zero-shot LOCALIZER[†] improves on NExT-QA by averaging 1.2%. Furthermore, our LOCALIZER refined on pseudo-labels outperforms fine-tuned ATP and Differentiable top-K, with a 2.2% average improvement across all question types. Overall, our LOCALIZER shows superior effectiveness in both settings.

Table 6: Comparison of our LOCALIZER with other keyframe localization methods.

| Method | NExT-QA | | | |
|---|---|---|---|---|
| | Tem. | Cau. | Des. | Avg. |
| ANSWERER | 59.7 | 60.8 | 73.7 | 62.4 |
| (zero-shot) | | | | |
| + CLIP [55] | 59.2 | 60.0 | 72.5 | 61.8 |
| + Moment-DETR [30] | 59.5 | 60.6 | 72.1 | 62.0 |
| + LOCALIZER[†] | **61.3** | **61.5** | **75.6** | **63.6** |
| (fine-tuning) | | | | |
| + ATP [3] | 60.4 | 61.3 | 73.4 | 62.8 |
| + Differentiable Top-K [8] | 59.5 | 59.7 | 72.7 | 61.6 |
| + LOCALIZER | **62.3** | **63.1** | **74.9** | **64.6** |

**Impact of keyframe selection ranges and quantities.** In Table 7, we evaluate temporal keyframe localization in a zero-shot setting using various keyframe selection ranges and quantities. Even with one keyframe selected, our LOCALIZER shows significant improvements over the BLIP-2$^\text{voting}$ based on majority vote on 8 frame-level answers: NExT-QA-Causal (+2.4%), NExT-QA-Description (+3.6%), and How2QA (+2.6%). This highlights our LOCALIZER's effectiveness in localizing selective keyframes. We also note that multiple keyframes benefit NExT-QA-Temporal questions, but denser frames result in worse performance. It supports our finding in Sec. 4.4, that using too dense frames may distract image-LM.

Table 7: Ablation of different numbers of input frames and output keyframes.

| Settings | NExT-QA | | | | How2QA |
|---|---|---|---|---|---|
| | Tem. | Cau. | Des. | Avg. | |
| BLIP-2$^\text{voting}$ (8) | 59.9 | 60.2 | 72.4 | 62.0 | 69.8 |
| 8→1 | 59.8 | 61.1 | **76.0** | 62.9 | 72.4 |
| 16→1 | 59.2 | **62.6** | 74.9 | 63.4 | **73.2** |
| 16→4 | 60.7 | 61.5 | 75.8 | 63.4 | 72.4 |
| 32→4 | **61.3** | 61.5 | 75.6 | **63.6** | 72.3 |
| 32→8 | 59.4 | 60.9 | 74.7 | 62.5 | 71.3 |
| 64→8 | 58.9 | 60.9 | 74.0 | 62.2 | 71.8 |

**Impact of different frame sampling during ANSWERER fine-tuning.** In Sec. 3.3, we discussed how the ANSWERER in the SEVILA framework can be further fine-tuned in the forward chain using keyframes from the LOCALIZER. Table 8 compares fine-tuning in different frame sampling strategies, and indicates SEVILA framework performs optimally when utilizing LOCALIZER in both ANSWERER training and evaluation. This is likely due to the provision of more informative keyframes and milder domain shifts between the training and evaluation.

**Upper-bound performance analysis on oracle keyframes.** We further explore the upper-bound performance with the assumption of a 'perfect' localizer, capable of always providing the right keyframes for the ANSWERER. To achieve this, we uniformly sample four frames from each video, input them into BLIP-2 individually, and generate four frame-level answers. The upper-bound performance is represented by the oracle accuracy, which considers a question correctly answered if at least one of the four frames yields the right answer. As shown in Table 9, significant gaps exist between BLIP-2 majority voting and oracle accuracy in both settings. These gaps emphasize the need for more future work in temporal localization to effectively harness image-LM for video-language tasks.

**Qualitative analysis on LOCALIZER.** In Fig. 4, we present an example from NExT-QA (more in Appendix), showcasing the QA pairs, our LOCALIZER results, and the ground truth task-related video clips that we manually annotated. Our LOCALIZER more accurately matches human annotations than

| Frame Sampling | | NExT-QA | | | |
|---|---|---|---|---|---|
| Training | Inference | Temp. | Cau. | Des. | Avg. |
| Random | Uniform | 68.1 | 72.9 | 81.2 | 72.6 |
| Random | Localizer[†] | 67.6 | **73.4** | **84.0** | 73.1 |
| Localizer[†] | Uniform | 68.2 | 72.7 | 80.0 | 72.3 |
| Localizer[†] | Localizer[†] | **68.8** | **73.4** | 83.5 | **73.4** |

Table 8: Comparing different frame sampling during ANSWERER fine-tuning. The LOCALIZER[†] is frozen during fine-tuning. We use 4 frames for ANSWERER training, while the LOCALIZER[†] is the default 32→4.

| Datasets | BLIP-2[voting] (Oracle) | |
|---|---|---|
| | Zero-Shot | Fine-tuned |
| NExT-QA (Avg.) | 62.7 (70.1) | 70.1 (79.7) |
| STAR (Avg.) | 40.3 (52.9) | 51.8 (72.2) |
| How2QA | 69.8 (77.8) | 79.6 (86.4) |
| TVQA | 35.7 (45.4) | 54.5 (69.0) |
| VLEP | 63.8 (70.5) | 67.0 (79.1) |

Table 9: BLIP-2[voting] and oracle (in brackets) performance analysis across datasets. We use 4 frames for each video question. **Oracle**: at least 1 of 4 frames can give the right answer.

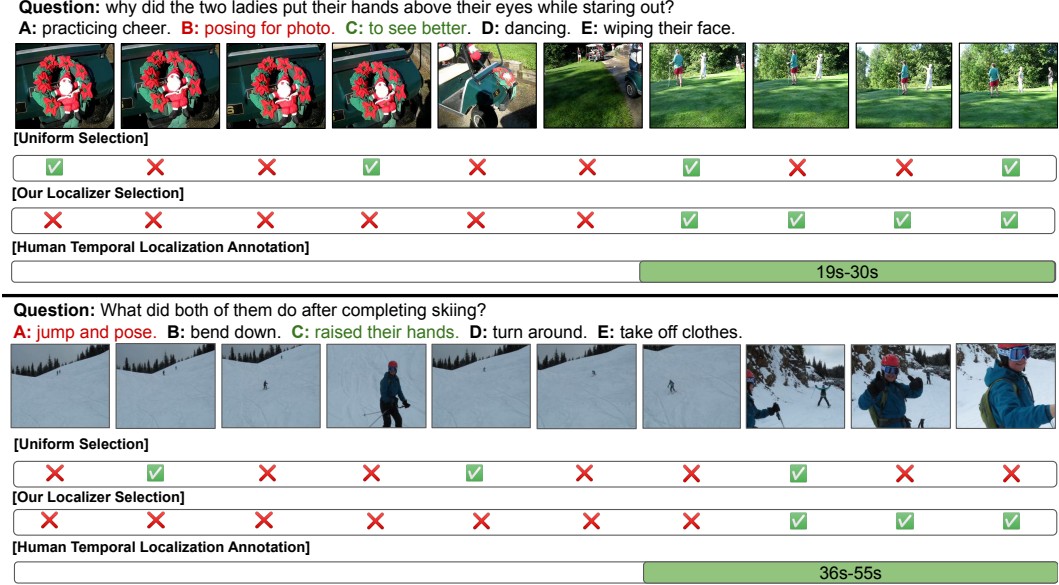

Figure 4: Visualization of our LOCALIZER. We use zero-shot ANSWERER with different frame sampling (uniform *v.s.* LOCALIZER) to answer the question. Red options are answered wrongly with uniformly sampled frames. Green options are answered correctly with our LOCALIZER. Best viewed in color.

uniform selection. This accurate localization enables the ANSWERER to correctly answer questions, while uniform selection leads to incorrect responses. These results demonstrate that our LOCALIZER can effectively locate task-related keyframes in a video, thus benefiting downstream tasks.

## 5 Conclusion and Future Work

In this paper, we introduced SEVILA, a novel video-language framework. SEVILA adapts an image-language model to obtain two modules: (1) LOCALIZER for language-aware temporal localization and (2) ANSWERER for question answering on keyframes. SEVILA tackles video-language tasks by chaining the output of LOCALIZER to the input of ANSWERER (*forward chain*), while ANSWERER can give feedback to refine the LOCALIZER (*backward chain*) via pseudo labeling. The proposed temporal localization allows a more focused understanding of videos and improves the accuracy of video-language tasks, while the pseudo-labeling process alleviates the need for expensive language-aware keyframe annotations. Compared with state-of-the-art baselines, SEVILA achieves competitive or better performance on five video questions answering and event prediction benchmarks. We also provide a comprehensive analysis elaborating on the design of the proposed two-stage self-chaining. Our research encourages future work to improve temporal localization in video understanding.

**Limitations & Broader Impacts.** See Appendix for limitations and broader impacts discussion.

## Acknowledgement

We thank the reviewers and Archiki Prasad, Hao Tan, Yi-Lin Sung, and Jie Lei for their valuable feedback and input for the paper. This work was supported by ARO Award W911NF2110220, DARPA KAIROS Grant FA8750-19-2-1004, DARPA MCS Grant N66001-19-2-4031, and NSF-AI Engage Institute DRL211263. The views, opinions, and/or findings contained in this article are those of the authors and not of the funding agency.

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

# Appendix

In this Appendix, we present the following:

- Additional comparison between image-language models (image-LMs) and video language models (video-LMs) in terms of model size and pre-training data size (Sec. A).

- Additional information about our experimental setup (Sec. B), including benchmarks details and task definitions (Sec. B.1), metrics (Sec. B.2), baseline implementation details (Sec. B.3), and SEVILA implementation details (Sec. B.4).

- Additional experiments (Sec. C), including the single-frame v.s. multi-frame LOCALIZER (Sec. C.1), iterative self-refinement results (Sec. C.2), different LOCALIZER pre-training settings (Sec. C.3), SEVILA framework with another Image-LM (MiniGPT4) (Sec. C.4), computational cost of SEVILA (Sec. C.5), the impact of prompt designs (Sec. C.6) and more qualitative visualization (Sec. C.7).

- Limitation and broader impact of this work (Sec. D), as well as license information (Sec. E) for the datasets, codes, and models that we used in this paper.

## A    Comparison Between Image-LMs and Video-LMs

In this section, we further provide a more visualized comparison between Image-LMs [55, 35, 43, 61, 70] and Video-LMs [71, 87, 37, 67, 17], in terms of model size and pre-training data size. As illustrated in Fig. 5, recent video-LMs generally have smaller scales in model and pre-training data sizes. This is because video-level annotations are more costly than image-level annotations, as discussed in the main paper Sec. 2 The gap between image-LMs and video-LMs has driven extensive research into image-to-video transfer learning to effectively leverage powerful image-LMs for video-language tasks. In this paper, we adopt this image-to-video transfer strategy by employing the state-of-the-art pre-trained image-LM, BLIP-2 [35].

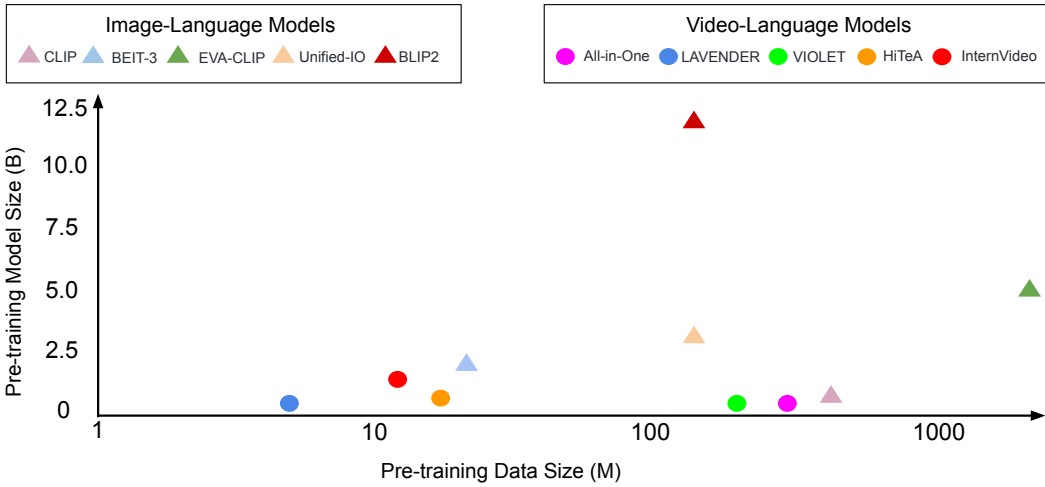

Figure 5: Comparison between image-LMs and video-LMs, in terms of model size and pre-training data size.

## B    Experimental Setup

In this section, we present additional information on the used datasets/benchmarks and task definitions (Sec. B.1), metrics (Sec. B.2), baseline implementations (Sec. B.3), and our SEVILA framework implementations (Sec. B.4).

| Dataset | # frames | Batch Size per GPU | Learning Rate | Epoch | Gradient Accumulation Step |
|---------|----------|--------------------|---------------|-------|----------------------------|
| NExT-QA | 8 | 4 | 1e-5 | 10 | 32 |
| STAR | 8 | 4 | 1e-5 | 10 | 32 |
| How2QA | 8 | 4 | 1e-5 | 10 | 32 |
| TVQA | 8 | 4 | 1e-5 | 10 | 32 |
| VLEP | 8 | 4 | 1e-5 | 10 | 32 |

Table 10: InternVideo fine-tuning hyperparameters.

## B.1  Benchmarks

**Video Question Answering (Video QA).** We test models on four challenging long-form multi-choice benchmarks: (1) **NExT-QA** [77], a benchmark for causal and temporal reasoning. It contains 5440 videos with an average length of 44s and approximately 52K questions. NExT-QA contains 3 different question types: Temporal (Tem.), Causal (Cau.), and Descriptive (Des.). (2) **STAR** [75], a benchmark for situated reasoning, which contains 22K video clips with an average length of 12s, along with 60K questions. STAR contains 4 different question types: Interaction (Int.), Sequence (Seq.), Prediction (Pre.), and Feasibility (Fea.). (3) **How2QA** [36], a benchmark containing 44k questions along with 22 thousand 60-second clips selected from 9035 videos. (4) **TVQA** [27], another benchmark contains 152K questions along with 21k video clips (76 seconds on average) from TV shows.

**Video Event Prediction (Video EP).** We test models on **VLEP** [28], a benchmark that requires the model to predict two future events based on the video premise. Thus, we can formulate this task as a multi-choice QA. VLEP contains 28,726 future event prediction cases from 10,234 diverse TV Shows and YouTube Lifestyle Vlog video clips.

**Video Moment Retrieval.** We test our LOCALIZER performance on **QVHighlights** [30], where a model predicts the temporal span in a video corresponding to a human-generated language description. QVHighlights consists of 10,148 videos with a duration of 150s, 18,367 moments, and 10,310 queries.

## B.2  Metrics

For video question answering and video event prediction, we adopt the standard answer accuracy. For video moment retrieval, we follow Lei et al. [30] and report the standard mean average precision (mAP) over multiple IoU thresholds [0.5: 0.05: 0.95], and the standard Recall@1 (R@1) where we define a prediction to be positive if it has a high IoU (>= 0.5 or >= 0.7) with one of the ground truth moments. For NExT-QA, STAR, How2QA, TVQA, and VLEP we report the performance on the validation set whereas QVHighlights we report on the hidden test set.

## B.3  Baseline Implementations

**Baseline Architecture**. For InternVideo [71], we adopt the released largest MM-L-14 variant that initialized from CLIP-L/14 [55] with 1B parameters and follow its default 8-frame setting for comparison. We fine-tuned InternVideo on downstream tasks by ourselves. For BLIP-2$^{voting}$ and BLIP-2$^{concat}$ (ANSWERER), we use the same BLIP-2 ViT-G Flan-T5 XL to our SEVILA framework.

**Baseline Training**. In Table 10, we show the setup of InternVideo [71] fine-tuning on downstream tasks. we adopt the released largest MM-L-14 variant with 1B parameters and follow its default 8-frame setting. We conduct experiments with 4 × 48GB A6000 GPUs. It takes about 4h on NExT-QA, 8h on STAR, 4h on How2QA, 12h on TVQA, and 2h on VLEP.

**Implementation Details of Other Keyframe Selection Methods.**. In the main paper Sec. 4.6, we compare our LOCALIZER with other keyframe selection methods in both zero-shot and fine-tuning settings. Specifically, in zero-shot setting, we compare our LOCALIZER with CLIP [55] and Moment-DETR [30], utilizing the same ANSWERER (32→4) for each. For CLIP one, we calculate the pre-trained CLIP-ViT-B/32 image-language similarity between each frame's visual feature and the corresponding question and option features. We then select the top-4 similarity keyframes from 32 uniformly sampled frames for the ANSWERER. In the case of Moment-DETR, we apply the Moment-DETR pre-trained on QVHighlights to first detect a temporal span corresponding to the question and option

| Dataset | Batch Size per GPU | Learning Rate | Warmup Step | Epoch | Gradient Accumulation Step |
|---|---|---|---|---|---|
| **LOCALIZER Pre-Training** | | | | | |
| QVHighlights | 64 | 3e-5 | 1000 | 80 | 1 |
| **ANSWERER Fine-tuning in Forward Chain** | | | | | |
| NExT-QA | 8 | 3e-5 | 1000 | 10 | 2 |
| STAR | 8 | 3e-5 | 1000 | 10 | 2 |
| How2QA | 4 | 3e-5 | 3000 | 10 | 4 |
| TVQA | 4 | 3e-5 | 8000 | 10 | 4 |
| VLEP | 4 | 1e-5 | 1200 | 10 | 4 |
| **LOCALIZER Self-Refinement in Reverse Chain** | | | | | |
| NExT-QA | 64 | 3e-5 | 500 | 10 | 1 |
| STAR | 64 | 3e-5 | 500 | 10 | 1 |
| How2QA | 64 | 3e-5 | 500 | 10 | 1 |
| TVQA | 64 | 3e-5 | 2000 | 10 | 1 |
| VLEP | 64 | 3e-5 | 500 | 10 | 1 |

Table 11: SEVILA framework training hyperparameters.

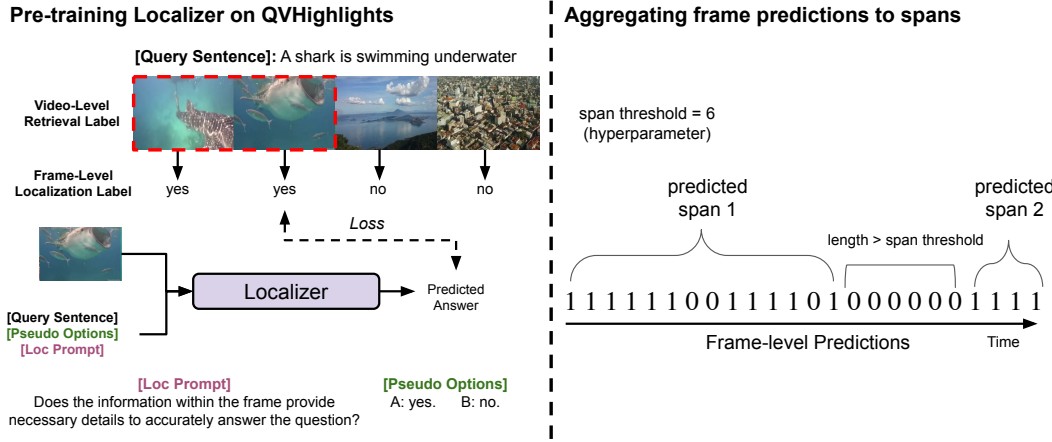

Figure 6: **Left:** For LOCALIZER pre-training, we utilize the video moment retrieval labels for the keyframe localization task. **Right:** we aggregate LOCALIZER's frame-level predictions into video-level span predictions.

sentence. Subsequently, we uniformly sample 4 frames within this span for the ANSWERER to predict answers. Next, we compare our LOCALIZER with the fine-tuned ATP and the Differentiable Top-K plugin. Both ATP [3] and Differentiable Top-K [8] utilize the BERT-base encoder [66], which is of the same size as the Q-former in our LOCALIZER. Both ATP and Differentiable Top-K modules are inserted after the Q-former to learn salient frame feature selection in an end-to-end manner.

### B.4 SEVILA Implementation Details

**SEVILA Architecture.** SEVILA framework adopts BLIP-2 [35], an image-language model with 4.1B parameters. We freeze its visual encoder (ViT-G [16]; 1B) and large language model (Flan-T5 XL [7]; 3B), and only fine-tune the Q-former (including its learnable queries), and a fully connected layer after Q-former for each of both LOCALIZER and ANSWERER. There are 106M trained parameters, which is 2.5% of the total parameters of BLIP-2.

**SEVILA Framework Training.** We conduct experiments with 4 × 48GB A6000 GPUs. For LOCALIZER per-training, we pre-train LOCALIZER on the QVHighlights for 80 epochs, taking approximately 12 hours with 4 × 29GB on A6000 GPUs. For LOCALIZER self-refinement, we train LOCALIZER on

| Answerer | # frames of Localizer | NExT-QA (Average) |
|---|---|---|
| zero-shot | 1 | **64.6** |
|  | 4 | 63.6 |
| fine-tuned | 1 | **73.4** |
|  | 4 | 71.3 |

Table 12: Comparison between single-frame and multi-frame Localizer.

each downstream dataset with pseudo-labels for an additional 10 epochs with $4 \times 29GB$ on A6000 GPUs. It takes about 7h on NExT-QA, 8h on STAR, 6h on How2QA, 17h on TVQA, and 3h on VLEP. For Answerer fine-tuning, we fine-tune Answerer on each downstream dataset with answer labels and frozen pre-trained Localizer for 10 epochs with $4 \times 48GB$ on A6000 GPUs. It takes about 13h on NExT-QA, 15h on STAR, 13h on How2QA, 48h on TVQA, and 8h on VLEP. We employ standard cross-entropy loss between the model outputs and the target values. We report SeViLA framework training hyperparameters in Localizer pre-training, Answerer fine-tuning, and Localizer self-refinment, in Table 11.

**Prompt Engineering.** We write multiple QA and Loc prompts shown in the main paper and choose the prompt yielding the best zero-shot performance on the downstream task (see Sec. C.6).

**Localizer Pre-training Details.** As illustrated in Fig. 6 (left), we provide the visualization on our Localizer pre-training. As discussed in the main paper Sec. 3.3, we adopt QVHighlighs [30] to pre-train our Localizer, we transfer its temporal span annotations to keyframe localization labels by comparing frame time-stamp with spans. We further design a prompt template and fill in it with query sentences in QVHighlighs. It makes the pre-training and downstream video QA and event prediction tasks with a similar input format and can further benefit downstream tasks.

**Details of Aggregation for Video Moment Retrieval.** We elaborate on the aggregation of frame-level predictions from our Localizer, as discussed in the main paper Sec. 4.5. As illustrated in Fig. 6 (right), we use a hyperparameter called 'span threshold', a maximum number of continuing '0's (continuing 'no' prediction for frame-level localization) inside a single span. If there are more '0's than the value in a row, we split the frames into separate spans. We use the span threshold=6. We determined this value based on statistics of QVHighlights training data. In QVHighlights, a query sentence has multiple grounding spans in a video. We calculate the average interval among those spans as our span threshold.

## C   Experiments

In this section, we present extra in-depth analysis on our Answerer and Localizer, including the single-frame v.s. multi-frame Localizer (Sec. C.1), iterative self-refinement results (Sec. C.2), different Localizer pre-training settings (Sec. C.3), SeViLA framework with another Image-LM (MiniGPT4) (Sec. C.4), computational cost of SeViLA (Sec. C.5), the impact of prompt designs (Sec. C.6) and more qualitative visualization (Sec. C.7)

### C.1   Single-frame v.s. Multi-frame Localizer

Even though our single-frame Localizer in SeViLA can locate language-aware keyframes, temporal modeling for video-level tasks cannot be overlooked. In this section, we've expanded our Localizer to a multi-frame mode by concatenating frames into a long image before Q-Former. This allows for full attention across all frames, enabling temporal modeling. As shown in Table 12 we find that the 4-frame Localizer performs worse than the single-frame Localizer in both zero-shot and fine-tuning settings. We suspect that this is because our backbone model (BLIP-2) has not seen video data during its pre-training. We think that a multi-frame Localizer could be more powerful once we have enough temporal grounding annotations or enough pre-training on large-scale video dataset [10]. We leave the improvement of the multi-frame Localizer to future work.

| Localizer | NeXT-QA (Average) |
|---|---|
| w/o Localizer | 62.4 |
| + Moment-DETR | 62.0 |
| + Our Localizer (without pre-training) | 62.9 |
| + Our Localizer (weakly pre-trained with QVH ASR) | 63.2 |
| + Our Localizer (pre-trained with QVH) | **63.6** |

Table 13: Comparison among different pre-training settings of Localizer.

| Iteration | NeXT-QA (Average) |
|---|---|
| 1 | 73.8 |
| 2 | **74.2** |
| 3 | 73.7 |

Table 14: Iterative self-refinement results of SeViLA framework.

## C.2 Iterative Self-refinement on Localizer and Answerer

Madaan et al. [48] introduces the 'Self-Refine' framework, where an LLM iteratively generates initial output, provides feedback, and refines the outputs. We experiment with iterative self-refinement, where the Answerer gives pseudo labels to train Localizer, and then the Localizer provides more accurate language-aware frames to fine-tune Answerer, iteratively. As shown in Table 14, we find that two iterations of self-refinement slightly increase performance compared to the single self-refinement, but the performance saturates from three iterations. We leave further analysis with self-refinement to future work.

## C.3 Different Pre-training Settings of Localizer

We explored weakly-supervised pre-training using ASR similar to Moment-DETR [30]. As shown in Table 13, our Localizer performance improves with the weakly supervised pre-training, closing the gap with the pre-training with manual annotations (QVHighlights [30]).

## C.4 SeViLA Framework with Another Image-LM (MiniGPT4)

We experimented with extending the SeViLA framework with another recent Image-Language model (MiniGPT4 [92]). On NeXT-QA, zero-shot MiniGPT4 Answerer achieves 52.7% average accuracy and obtains 0.7% boost with zero-shot MiniGPT4 Localizer. We find our proposed self-chaining scheme can also improve the performance of zero-shot MiniGPT4, and we expect that fine-tuning the model will even improve the performance.

| Model | Memory (GB) | Running Time (sec./sample) | Parameter (B) |
|---|---|---|---|
| Answerer (4) | 7.56 | 1.79 | 4.1 |
| SeViLA (32->4) | 7.98 | 3.28 | 4.2 |

Table 15: Computational cost of SeViLA framework.

## C.5 Analysis on Computational Cost of SeViLA Framework

In Table 15, we show memory, running time, and parameter size (ViT + FlanT5 + Qformer) comparison between our Answerer and our SeViLA (Localizer+Answerer). As the Localizer and Answerer share the most parameters, adding Localizer has a very small additional memory footprint.

## C.6 Impact of Prompt Design

We write several localization prompts and choose the one giving the best performance. We examine the impact of various localization prompts as shown in Table 16. For this experiment, we employ a

| Localization Prompt | NExT-QA | | | |
|---|---|---|---|---|
| | Temporal | Casual | Descriptive | Average |
| Does the frame have the information needed to answer the question correctly? | 59.9 | 61.1 | 74.2 | 62.7 |
| Does the provided frame contain the necessary information to accurately answer the given question? | 59.9 | 60.8 | 75.0 | 62.7 |
| Does the information within the frame provide the necessary details to accurately answer the given question? | 60.4 | 61.0 | 74.6 | 62.9 |

Table 16: Impact of different localization prompts on the zero-shot Video QA performance

zero-shot ANSWERER and an untrained LOCALIZER to choose the most effective localization prompt based on performance. It demonstrates that our model is insensitive to the changes in the prompt.

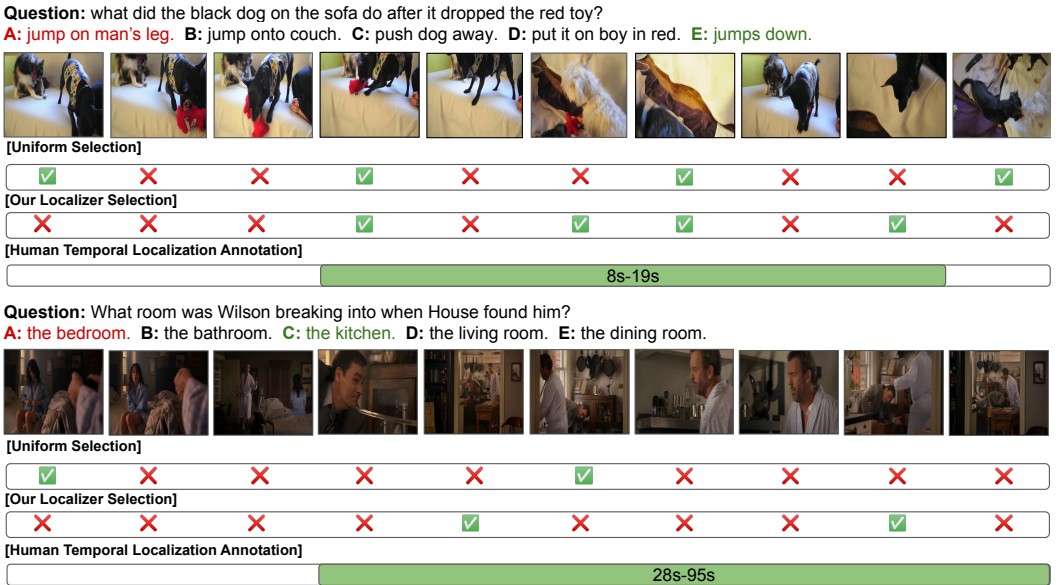

Figure 7: Visualization of our LOCALIZER. We show various keyframe amounts in those examples. We use zero-shot ANSWERER with different frame sampling (uniform *v.s.* LOCALIZER) to answer the question. Red options are answered wrongly with uniformly sampled frames. Green options are answered correctly with our LOCALIZER. Best viewed in color.

## C.7 Visualization

As illustrated in Fig. 7, we provide more visualization examples from different datasets, and with various selected keyframe amounts. Compared with uniform sampling, our LOCALIZER can locate relevant frames that match human annotations well regardless of keyframe amounts. And such a good keyframe localization also brings the correct answers to corresponding questions.

## D   Limitations and Broader Impacts

**Limitations.**   Although our SEVILA framework can help locate language-aware keyframes in many real-world video tasks, *e.g.*, video content analysis and event detection, our LOCALIZER uses frame-level keyframe localization and might not handle well some complex, fine-grained temporal events (*e.g.*, open the door *vs.* close the door). Future work can explore structured prediction for temporal localization beyond the frame level.

**Broader impacts.** SᴇVɪLA framework leverages a large image-language model trained on massive internet-scale data. Similar to most such large models, this might occasionally yield unexpected or inappropriate responses, potentially reflecting societal biases related to gender, race, or sexuality [47]. More studies in the large image-language model are needed to evaluate and mitigate these negative biases, and toxic output.

# E  License

We will make our code and models publicly accessible. We use standard licenses from the community and provide the following links to the licenses for the datasets, codes, and models that we used in this paper. For further information, please refer to the specific link.

**NExT-QA [77]:** MIT

**STAR [75]:** Apache

**How2QA [36]:** MIT

**TVQA [27]:** MIT

**VLEP [28]:** MIT

**QVHighlights [30]:** CC BY-NC-SA 4.0

**LAVIS [33]:** BSD 3-Clause

**PyTorch [53]:** BSD-style

**Huggingface Transformers [74]:** Apache

**Torchvision [49]:** BSD 3-Clause

