# OpenReview forum: "Self-Chained Image-Language Model for Video Localization and Question Answering"
_NeurIPS.cc/2023/Conference — NeurIPS 2023 poster_

### Official Review · Reviewer_mWtS · 2023-06-17

**Soundness:** 3 good
**Presentation:** 3 good
**Contribution:** 3 good
**Rating:** 6
**Confidence:** 4

**Summary:**

This work builds a joint model for temporal language grounding and video question answering. The model is built on a state-of-the-art image-language model, BLIP-2, and finetunes it in a parameter-efficient way to derive a localizer and an answerer. The localizer finds language-aware keyframes in a video and the answerer uses these frames to predict the answer. The answerer is also used to generate pseudo-labels to refine the localizer. Experiments on multiple video question answering datasets in both finetuning and zero-shot settings show that the proposed method achieves state-of-the-art results and improves slightly over the BLIP-2 baseline.


**Strengths:**

- Leveraging image-language models for video-language tasks is an important direction, and localizing language-aware keyframes seems a good idea to achieve this.
- The method achieves strong results on a wide variety of benchmarks in both finetuning and zero-shot mode.


**Weaknesses:**

- Most of the improvement compared to state-of-the-art methods come from the use of the BLIP-2 model. The proposed method only show small improvement over BLIP-2+concat in finetuning setting (see Table 1).
- Can the method generalize with other image-language models, e.g. BLIP, and if so how does it perform?
- The localizer relies on manual temporal language grounding annotations from QVHighlights. Could it benefit from weak supervision from e.g. HowTo100M like Moment-DETR?


**Questions:**

See weaknesses.

**Limitations:**

Limitations and potential negative societal impact are discussed in the Supplementary Material.

---

> ### Author Rebuttal · Authors · 2023-08-10
>
> ### **Weakness 1**
> > Most of the improvement compared to state-of-the-art methods comes from the use of the BLIP-2 model. The proposed method only shows small improvement over BLIP-2+concat in fine-tuning setting (see Table 1).
>
> &nbsp; Our SeViLA consists of Localizer + Answerer, where both have BLIP-2 (visual encoder + Q-former + LLM) architecture, with separate Q-former parameters. Note that the ‘**BLIP2 concat**’ in Tables 1&2 is equivalent to our '**Answerer**' as a contribution that extends BLIP-2 to adapt to video.
>
> &nbsp; Our Answerer (BLIP-2 concat) performs better than the original BLIP-2 with single frame inputs (BLIP-2 voting). And our SeViLA which incorporates our Localizer with our Answerer further boosts the performance. In both fine-tuning and zero-shot settings (Tables 1&2, and as follows), our SeViLA outperforms the original BLIP-2 (BLIP-2-voting) with non-trivial improvements.
>
> | **Model** | **NeXT-QA (Avg.)** | **STAR (Avg.)** | **How2QA** | **TVQA** |
> |----|----|----|----|----|
> | (Zero-shot) | | | | |
> | BLIP-2 voting | 62.7 | 40.3 | 69.8 | 35.7 |
> | SeViLA | 63.6 (+0.9) | 44.6 (+4.3) | 72.3 (+2.5) | 38.2 (+2.5) |
> | (Fine-tuning) | | | | |
> | BLIP-2 voting | 70.1 | 51.8 | 79.6 | 54.5 |
> | SeViLA | 73.8 (+3.7) | 64.9 (+13.1) | 83.6 (+4.0) | 61.6 (+7.1) |
>
>
> ### **Weakness 2**
> > Can the method generalize with other image-language models, e.g. BLIP, and if so how does it perform?
>
>
> &nbsp; As you suggested, we experimented with extending SeViLA with another recent Image-Language model (MiniGPT4 [a]), and show the results as follows. We find our proposed self-chaining scheme can also improve the performance of zero-shot MiniGPT4, and we expect that finetuning the model will even improve the performance.
>
> | **Model** | **NExT-QA (Avg.)** |
> |----------------------------------------|--------------------|
> | MiniGPT4 Answerer | 52.7 |
> | MiniGPT4 Localizer + MiniGPT4 Answerer | 53.5 |
>
> Ref:
> [a] D. Zhu et.al. MiniGPT-4: Enhancing Vision-language Understanding with Advanced Large Language Models. ArXiv pre-print, 2023.
>
> ### **Weakness 3**
> > The localizer relies on manual temporal language grounding annotations from QVHighlights. Could it benefit from weak supervision from e.g. HowTo100M like Moment-DETR?
>
> &nbsp; We’d like to note that our localizer brings 0.5% (62.4 in Table 3 row B v.s. 62.9 in Table 5) boost on NeXT-QA, without pretraining. Our localizer even outperforms the Moment-DETR pre-trained on HowTo100M + QVH in Table 6 (comparison with other localization methods).
> This shows that our temporal grounding method is already effective even without manual annotations.
>
> &nbsp; Following your suggestion, we also explored weakly-supervised pretraining using ASR similar to Moment-DETR. As shown in the following table, our Localizer performance improves with the weakly supervised pretraining, closing the gap with the pretraining with manual annotations (QVH).
>
> | **Localizer** | **NeXT-QA (Avg.)** |
> |--------------------------------------|--------------------|
> | w/o Localizer | 62.4 |
> | Moment-DETR | 62.0 |
> | Our Localizer (without pre-training) | 62.9 |
> | Our Localizer (weakly pre-trained with QVH ASR) | 63.2 |
> | Our Localizer (pre-trained with QVH) | 63.6 |

---

> > ### Comment · Reviewer_mWtS · 2023-08-10
> >
> > I thank the authors for providing a rebuttal, have read the other reviews, and stand by my original rating of weak accept. Below are detailed answers:
> >
> > Weakness 1: I agree with the authors that adapting BLIP-2 to videos by concatenating visual features output from the Q-former outperforms applying BLIP-2 separately to all video frames and using voting. However, I do not see this as a major contribution given that concatenating visual features from different frames before multi-modal modeling is a standard technique used in video-text modeling, see MERLOT for instance.
> >
> > Weakness 2: I appreciate seeing that MiniGPT-4 also benefits from the localizer for zero-shot NeXT-QA, and encourage the authors to include additional image-text models and/or other evaluation datasets to confirm the generalizability of the proposed localizer+answerer approach.
> >
> > Weakness 3: I appreciate seeing that the NeXT-QA accuracy improves with the weakly-supervised pretraining of the localizer on ASR.

---

> > > ### Author Response · Authors · 2023-08-21
> > >
> > > Thank you for your thoughtful feedback and we are glad that you appreciate MiniGPT-4 and ASR results on NeXT-QA. We agree that concatenating visual features isn’t our primary contribution (although one interesting difference is that we do not require fine-tuning). We will add all these changes and clarifications to the final version.

---

### Official Review · Reviewer_wzBa · 2023-07-06

**Soundness:** 3 good
**Presentation:** 3 good
**Contribution:** 2 fair
**Rating:** 5
**Confidence:** 4

**Summary:**

The paper presents the SeViLA framework, which addresses the limitations of existing image-language models for video question answering. It introduces two modules, Localizer and Answerer, that are fine-tuned from a pre-trained image-language model. SeViLA utilizes chaining for cascaded inference and self-refinement. The Localizer identifies language-aware keyframes, and the Answerer predicts the answer based on these keyframes. Additionally, the Answerer generates keyframe pseudo-labels to refine the Localizer, eliminating the need for costly video moment annotations.

**Strengths:**

1.	The paper is well-written and easy to understand.
2.	The concept of localizing and then answering is intuitive and effective.
3.	Extensive experiments demonstrate that SeViLA achieves state-of-the-art performance on challenging video question answering benchmarks. The paper also provides a comprehensive analysis, including comparisons with other localization models and the impact of keyframe variations.


**Weaknesses:**

1.	The motivation and LGDN (which also addresses VideoQA tasks) are quite similar. While the paper mentions LGDN in the introduction and related work sections (which I appreciate), I suggest the authors further clarify the differences between the two approaches in the related work section.
2.	The self-chaining approach requires additional computational resources (key frames need to be recomputed). The paper should provide information on the additional computational costs involved.


**Questions:**

Please see weaknesses.

**Limitations:**

Please see weaknesses.

---

> ### Author Rebuttal · Authors · 2023-08-10
>
> ### **Weakness 1**
>
> > The motivation and LGDN are quite similar. While the paper mentions LGDN in the introduction and related work sections (which I appreciate), I suggest the authors further clarify the differences between the two approaches.
>
> &nbsp; While LDGN selects salient frames with two different modules an image model (ViT) and language model (BERT), and answers questions by a QA model (Transformer), with multiple training objectives, our SeViLA proposes a simple and unified self-chaining of a single image-language model (BLIP-2) to tackle both (1) keyframe localization task and (2) question answering task. We will add more clarification of this in the paper.
>
> &nbsp; As pointed out by Reviewer mWtS, leveraging of image-language model for video-language tasks is an important further direction, and as recognized by Reviewer TABX and b5LJ, our work shows a novel self-chained way to effectively use a single large image-language model for video-language tasks.
>
> ### **Weakness 2**
> > The self-chaining approach requires additional computational resources (key frames need to be recomputed). The paper should provide information on the additional computational costs involved.
>
> &nbsp; In the following table, we show memory, running time, and parameter size (ViT + FlanT5 + Qformer) comparison between our Answerer and our Localizer+Answerer. As the Localizer and Answerer share the most parameters, adding Localizer has a very small additional memory footprint.
>
> | **Model** | **Memory (GB)** | **Running Time (sec./sample)** | **Parameter (B)** |
> |----|--|--------------|-------------------|
> | Answerer (4) | 7.56 | 1.79 | 4.1 |
> | Localizer + Answerer (SeViLA, 32->4) | 7.98 |  3.28 | 4.2 |

---

> > ### Comment · Reviewer_wzBa · 2023-08-18
> >
> > Thank you for your response! Most of my concerns are solved, so I would like to maintain my positive score.

---

> > > ### Author Response · Authors · 2023-08-21
> > >
> > > Thanks for your response and we are glad that our response has addressed your questions and that you have a positive rating!

---

### Official Review · Reviewer_b5LJ · 2023-07-10

**Soundness:** 3 good
**Presentation:** 4 excellent
**Contribution:** 3 good
**Rating:** 6
**Confidence:** 4

**Summary:**

The paper proposes a simple yet effective framework for video localization and question answering using pretrained image-language models. The key idea is to introduce a LOCALIZER module which localizes the keyframes in a video in order to ignore irrelevant frames and better answer the question. The LOCALIZER module is realized with a separate image-language model and the module can be further improved by fine-tuning on the pseudo labels generated by the QA results from the ANSWER module. The proposed framework, SEVILA, achieves state-of-the-art VQA results on multiple challenging benchmarks under both fine-tuned and zero-shot setups.

**Strengths:**

1. The proposed framework is simple yet effective in extending image-language models to tackle video-related tasks such as video question answering and moment localization. The idea of using the LOCALIZER module to detect keyframes in a long video is technically sound and the results are convincing.

2. The paper provides extensive ablation study to analyze different components of the framework under different setups. E.g., the impact of selecting keyframes for VQA, the impact of self-refinement for the LOCALIZER module, and how the LOCALIZER compared with some existing keyframe localization methods. While there exists many keyframe localization methods, it's interesting to see the language-model-based method used in this paper gets comparable or even better results in the ablation study.

3. The paper is well organized, clearly written and the description of technical details is clear and easy to follow.

**Weaknesses:**

1. The technical contribution of the work is relatively weak. The idea of selecting keyframes in a video for VQA is straightforward and widely explored in prior work. The proposed self-refinement strategy is also a standard semi-supervised learning approach and there's no new training strategies / techniques proposed for the fine-tuning of image-language models.

2. While I understand the main goal of this work is to utilize image-language models for VQA, the limitation of image-based models for video-related tasks should not be ignored. First, image-based models may not be sufficient to accurately localize the keyframes in a long video. Second, temporal modeling is also missing in the ANSWERER module, which makes the model less capable of modeling temporal dependencies compared with video-based models. These limitations may not be significant in the current VQA benchmarks since the questions are less sensitive to temporal dependencies and reasoning.

**Questions:**

1. Is it necessary to use two different Q-former for the LOCALIZER and ANSWERER? Since both models are image based, what if we share the same Q-former for both modules?

2. What if we make the self-refinement an iterative process? In other words, we refine the ANSWERER module after the self-refinement of LOCALIZER, and vice versa. It's interesting to see when the performance would saturate with such interactive refinement process.

A minor suggestion: I don't think Figure 4 is a good qualitative example because (1) the action of the ladies is barely visible in the small figure, let alone the intention of the action; (2) the question can be answered by only checking the question itself without knowing the visual content, since "staring out" and "hands above their eyes" are explicated mentioned.

**Limitations:**

The authors have discussed the limitations of the work and its potential social impact.

---

> ### Author Rebuttal · Authors · 2023-08-10
>
> ### **Weakness 1**
> > The technical contribution of the work is relatively weak. The idea of selecting keyframes in a video for VQA is straightforward and widely explored in prior work. The proposed self-refinement strategy is also a standard semi-supervised learning approach and there's no new training strategies / techniques proposed for the fine-tuning of image-language models.
>
> &nbsp; We agree that keyframe selection in VideoQA is a popular topic as we mentioned in the related works (Line 94-102).
> Different from previous works [23, 40, 49] which link an image model (e.g., ViT) and a language model (e.g., BERT) to first select keyframes by vision & text similarity and then answer questions with a third QA model, we self-chained a single image-langue model to tackle both keyframe localization and question answering by prompting LLM. Our SeViLA shows the pre-trained large image-language model with Parameter Efficient Fine-Tuning (PEFT) is able to handle both video localization and question answering in a self-chaining style.
>
> &nbsp; As Reviewer mWtS pointed out, the image-LM for video-language tasks is an important direction since the image-LMs are improving rapidly, and our work shows an efficient way to leverage those image LMs for video-level tasks. Besides, as you mentioned, our work shows that image LM has a strong potential for video localization (Table 4 video moment retrieval result & Table 6 comparison with other keyframe localization methods).
>
> ### **Weakness 2**
>
> > First, image-based models may not be sufficient to accurately localize the keyframes in a long video.
>
> &nbsp; We’d like to note that multiple recent works [3,31] have also found the effectiveness and efficiency of Single-frame localizer in various long video QA benchmarks. In our experiment, as listed in Table 3 (and the below table), our single-frame Localizer demonstrates non-trivial performance boosts on a multitude of long video QA benchmarks.
>
> | **Model** | **NeXT-QA (Avg.)** | **STAR (Avg.)** | **How2QA** | **TVQA** |
> |---|---|----|---|---|
> | (zero-shot) | | | | |
> | Answerer-only | 62.4 | 42.2 | 70.8 | 36.6 |
> | SeViLA | 64.6 (+2.2) | 45.5 (+3.3) | 72.9 (+2.1) | 39.1 (+2.5) |
> | (fine-tuning) | | | | |
> | Answerer-only | 72.6 | 62 | 82.2 | 59.8 |
> | SeViLA | 73.8 (+1.2) | 64.9 (+2.9) | 83.6 (+1.4) | 61.6 (+1.8) |
>
> > the limitation of image-based models for video-related tasks should not be ignored.
>
> &nbsp; We acknowledge that the significance of temporal modeling for video-level tasks cannot be overlooked. In various sections, we emphasize its role in video-language tasks and suggest enhancements in temporal designs for the Localizer. In the Limitation (supp Line 136-139), we also point out the shortcoming and failure cases of the current single-frame Localizer. In Section 4.5 (Line 248-253) and Table 4 (results on video moment retrieval), we suggest more temporal designs for the Localizer to improve language-aware localization. Moreover, our analysis in Section 4.6 (Line 294-299), Table 7 (oracle performance analysis) and the Limitation (supp Line 136-139) indicates potential areas of improvement and underscores the importance of future research in temporal localization.
>
> &nbsp; Furthermore, in this rebuttal, we've expanded our Localizer to a multi-frame mode by concatenating frames into a long image **before** Q-Former. This allows for full attention across all frames, enabling temporal modeling. The results are the following, we find that the 4-frame localizer performs worse than the single-frame localizer in both zero-shot and finetuning settings.
>
> | **Answerer** | **# frames of Localizer** | **NExT-QA (Avg.)** |
> |----|----|----|
> | zero-shot | 1 | 64.6 |
> | zero-shot | 4 | 63.6 (-1.0) |
> | fine-tuned | 1 | 73.4 |
> | fine-tuned | 4 | 71.3 (-2.1) |
>
> > Second, temporal modeling is also missing in the ANSWERER module, which makes the model less capable of modeling temporal dependencies compared with video-based models.
>
> Please see the general response.
>
> ### **Question 1**
> > What if we share the same Q-former for both modules?
>
> &nbsp; Please note that Table 5 row 1 shows the setting where both Localizer and Answerer share the entire parameters, including Q-former. The model achieves 62.9% on NeXT-QA and 70.7% on How2QA, outperforming strong InternVideo and BLIP-2 voting baselines. We will also experiment with multi-task finetuning and discuss the results in the paper.
>
> ### **Question 2**
> > What if we make the self-refinement an iterative process?
>
> &nbsp; This is an interesting idea, thanks for your suggestion! We conducted experiments of iterative self-refinement and show the results in the table below. We found that self-refinement helps until two iterations and saturates afterward. We will incorporate this result in the paper.
>
> | **Iteration** | **NExT-QA (Avg.)** |
> |-----------|--------------------|
> | 1 (current) | 73.8 |
> | 2 | 74.2 |
> | 3 | 73.7 |
>
> ### **Minor Suggestion**
> > Qualitative visualization example
>
> &nbsp; Thanks for your suggestion on the qualitative example, we argue that even in this example visual content is necessary, because the correct answer "to see better" and not "to pose for photo" is only clear once the video shows that they are playing golf or there is no photographer.
>
> &nbsp; Please note that we have additional visualization examples in the supplementary Figure 3, where option events are all possible to happen without seeing videos and our model localizes the correct moment and gives the right answer.
> ```
> Q: What did both of them do after completing skill?
> A1: jump A2: bend down A3: raise their hands A4: turn around A5: take off clothes
> ```
>
> We will update the qualitative example in the main paper with the new example in the rebuttal PDF to show the effectiveness of our method better.

---

> > ### Comment · Reviewer_b5LJ · 2023-08-21
> >
> > Thanks for your response. Most of my concerns are well addressed in the rebuttal and I'd keep my original positive rating.

---

> > > ### Author Response · Authors · 2023-08-21
> > >
> > > Thanks for your response and we are glad that our response has addressed your questions and that you have a positive rating!

---

### Official Review · Reviewer_TABX · 2023-07-11

**Soundness:** 4 excellent
**Presentation:** 3 good
**Contribution:** 3 good
**Rating:** 5
**Confidence:** 4

**Summary:**

This paper proposes a novel framework called Self-Chained Video Localization-Answering (SEVILA) that leverages a single image-language model (BLIP-2) for both temporal keyframe localization and question answering in videos. The framework consists of two modules, Localizer and Answerer, which are parameter-fine-tuned (LoRA finetuning) from BLIP-2. The paper demonstrates that the SEVILA framework outperforms several strong previous works in video question answering and event prediction benchmarks.

**Strengths:**

(1) The paper proposes a novel framework that addresses the issue of missing important visual cues in video question answering tasks by leveraging a single image-language model for keyframe localization and question answering.
(2) The self-chaining mechanism in the framework allows for cascaded inference and self-refinement, improving the accuracy of temporal localization without the need for expensive annotations.
(3) The paper provides a comprehensive analysis of the proposed framework, including the impact of temporal localization, the self-refinement process, and the number of keyframes.

**Weaknesses:**

1. In the backward chain, the pseudo-tags are frame-level, and if a single frame of information can not give the correct answer, the Localizer is considered to have given the wrong keyframe. This is obviously unreasonable and ignores the important time dependencies provided by the intermediate frame. This is probably why there was no significant change in their performance when the self-refine part was removed in the ablation study.
2. the proposal is not good at dealing with temporal causality. Because Blip-2 is a image-text MLLM model, there is no space-time inference, and their answer is to directly add the q-former query generated by each frame to the word token of LLM. This is equivalent to letting q-former himself to realize the time level of information understanding, and q-former does not have this ability.
Could you make me clear?

**Questions:**

see above.

**Limitations:**

see above.

---

> ### Author Rebuttal · Authors · 2023-08-09
>
> ### **Weakness 1**
>
> > In the backward chain, the pseudo-tags are frame-level, and if a single frame of information can not give the correct answer, the Localizer is considered to have given the wrong keyframe.
>
> &nbsp; We note that even allowing multi-frame in Localizer can also cause the misleading issue in the single-frame Localizer, since both one frame and short video clips have a chance to give incorrect pseudo labels. Because there is no frame-level annotation, this is a general issue for such a weakly supervised pseudo-label training scheme despite single or multiple frame(s).
>
> > This is obviously unreasonable and ignores the important time dependencies provided by the intermediate frame.
>
> &nbsp; **We still acknowledge the importance of temporal modeling** for video-level tasks.
> In Section 4.2.a (Line 190-193), we highlighted that temporal modeling matters for video-language tasks. In Section 4.5 (Line 248-253) and Table 4 (results on video moment retrieval), we suggest more temporal designs for the Localizer to improve language-aware localization. In Section 4.6 (Line 294-299) and Table 7 (oracle performance analysis), we show that better localizers further improve the video question answering accuracy, emphasizing the need for more future work in such temporal localization. In the appendix Sec. D Limitation (appendix Line 136-139), we also discuss the failure cases of the current single-frame Localizer.
>
> > This is probably why there was no significant change in their performance when the self-refine part was removed in the ablation study.
>
> &nbsp; Our single-frame Localizer can improve video QA performance even without self-refinement.
> As shown in Table 3 rows B, C, and D, our single-frame Localizer demonstrates non-trivial performance boosts on a multitude of long video QA benchmarks.
>
> | **Model** | **NeXT-QA (Avg.)** | **STAR (Avg.)** | **How2QA** | **TVQA** |
> |-------------------------------|----------------|-------------|-------------|-------------|
> | B - Answerer-only | 62.4 | 42.2 | 70.8 | 36.6 |
> | C - SeViLA+ (Localizer + Answerer) | 63.6 (+1.2) | 44.6 (+2.4) | 72.3 (+1.5) | 38.2 (+1.6) |
> | D - SeViLA (Localizer + Answerer + self-refinement) | 64.6 (+**2.2**) | 45.5 (+**3.3**) | 72.9 (+**2.1**) | 39.1 (+**2.5**) |
>
> &nbsp; Inspired by your suggestion, we also explored using a multi-frame localizer, but found that the single-frame Localizer works better than the multi-frame Localizer. As shown in the table below, we find that the 4-frame localizer performs worse than the single-frame localizer in both zero-shot and finetuning settings. We suspect that this is because our backbone model (BLIP-2) has not seen video data during its pretraining. We think that multi-frame Localizer could be more powerful once we have enough temporal grounding annotations. We leave the improvement of the multi-frame Localizer to future work.
>
> | **Answerer** | **# frames of Localizer** | **NExT-QA (Avg.)** |
> |----|------------------------|--------------------|
> | zero-shot | 1 | 64.6 |
> | zero-shot | 4 | 63.6 (-1.0) |
> | fine-tuned | 1 | 73.4 |
> | fine-tuned | 4 | 71.3 (-2.1) |
>
> ### **Weakness 2**
> > The proposal is not good at dealing with temporal causality. Because Blip-2 is an image-text MLLM model, there is no space-time inference, and their answer is to directly add the q-former query generated by each frame to the word token of LLM. This is equivalent to letting q-former himself to realize the time level of information understanding, and q-former does not have this ability. Could you make me clear?
>
> Please see the general response.

---

### Author Rebuttal · Authors · 2023-08-09

We thank the reviewers for their time and valuable comments. We appreciate that reviewers recognized:
- motivation of our localization+answering design (b5LJ, wzBa)
- novelty of our self-chaining framework design (TABX)
- strong experimental results (TABX, b5LJ, wzBa, mWtS)
- extensive ablation studies and analysis (TABX, b5LJ, wzBa)
- well-organized presentation, clear description (b5LJ, wzBa)
- contribution in extending the image-language model for video-language tasks (b5LJ, mWtS).

In the responses, we include the following discussion/experiments.
- temporal modeling in both Localizer and Answerer (TABX, b5LJ)
- our novelty compared with prior work (wzBa)
- exp1: single-frame v.s. multi-frame Localizers (TABX)
- exp2: Single model version - Localizer and Answerer sharing Q-former (b5LJ)
- exp3: iterative self-refinement (b5LJ)
- exp4: SeViLA with another Image-LM (MiniGPT4) (mWtS)
- exp5: weakly supervised pre-training of Localizer (mWtS)

**Please also find the attached PDF for an additional qualitative example (Reviewer b5Lj).**

Below we answer to a common question about **Temporal modeling in Answerer** by Reviewer TABX W2 and Reviewer b5LJ W2

Reviewer TABX W2:
> The proposal is not good at dealing with temporal causality. Because Blip-2 is an image-text MLLM model, there is no space-time inference, and their answer is to directly add the q-former query generated by each frame to the word token of LLM. This is equivalent to letting q-former himself to realize the time level of information understanding, and q-former does not have this ability. Could you make me clear?

Reviewer b5LJ W2:
> Second, temporal modeling is also missing in the Answerer module, which makes the model less capable of modeling temporal dependencies compared with video-based models. These limitations may not be significant in the current VQA benchmarks since the questions are less sensitive to temporal dependencies and reasoning.

&nbsp; Our Answerer conducts temporal modeling with multiple frames via LLM rather than Q-former. As shown in Figure 2 and Line 147-149, Answerer Q-former does not take multiple frames at once but processes them into query features independently, and then those features are concatenated in a temporal order to feed into LLM with Frame ID hints. (Note that the ‘**BLIP2 concat**’ in Tables 1&2 is equivalent to our '**Answerer**' as a contribution that extends BLIP-2 to adapt to video).

&nbsp; According to fine-tuning results in Table 1 (and in the table below), we find that LLM in our Answerer has the capabilities of temporal understanding with multiple frame inputs, since our Answerer performs better than the BLIP-2 voting, which outputs answers by majority voting with the original BLIP-2 taking a single frame input:

| **Model** | **NeXT-QA (Avg.)** | **STAR (Avg.)** | **How2QA** | **TVQA** |
|----|----|----|----|----|
| BLIP-2 voting | 70.1 | 51.8 | 79.6 | 54.5 |
| Our Answerer | 72.6 (+**2.5**) | 62.0 (+**10.2**) | 82.2 (+**2.6**) | 59.8 (+**5.3**) |

&nbsp; Such a temporal understanding ability via LLM is evident in the case of the STAR-Sequence, a task demanding heavy temporal understanding, our Answerer outperforms BLIP-2 voting by a significant **13.1%** (fine-tuning Table 1) and **5.3%** (zero-shot Table 2). Those observations are also highlighted in Section 4.2.a where we discussed the importance of temporal modeling (Line 190-193).

---

### Comment · Area_Chair_kEmK · 2023-08-17
**Please respond to authors' rebuttal**

Dear Reviewers,

Thanks for your contributions! Please respond to the authors after carefully reading their rebuttals and other reviews. If your assessment of the paper changes, please update your score with a short justification for the new rating.

Thank you,

AC

---

### Decision · Program_Chairs · 2023-09-21

**Decision:**

Accept (poster)

**Comment:**

This paper has 4 positive ratings: 2 Borderline Accepts, and 2 Week Accepts. Before the rebuttal, the main concerns about this work are: 1) The novelty is limited compared to previous work. 2) The temporal modeling ability of the proposed method is limited. 3) More comprehensive experimental results are needed. After the rebuttal, all reviewers think most of their concerns have been addressed, and keep a positive rating for this paper. In summary, I recommend accepting this paper, and hope the authors can incorporate all these discussions into the final camera-ready version.